# Genetic Association of Beta-Myosin Heavy-Chain Gene (MYH7) with Cardiac Dysfunction

**DOI:** 10.3390/genes13091554

**Published:** 2022-08-29

**Authors:** Memoona Yousaf, Waqas Ahmed Khan, Khurrum Shahzad, Haq Nawaz Khan, Basharat Ali, Misbah Hussain, Fazli Rabbi Awan, Hamid Mustafa, Farah Nadia Sheikh

**Affiliations:** 1Department of Biotechnology, University of Sargodha, Sargodha 40100, Pakistan; 2Institute of Clinical Chemistry, University Hospital Leipzig Institute of Clinical Chemistry Liebigstraße27, D-04103 Leipzig, Germany; 3Diabetes and Cardio-Metabolic Disorders Laboratory, Health Biotechnology Division, National Institute for Biotechnology and Genetic Engineering (NIBGE), Faisalabad 38000, Pakistan; 4Department of Family Medicine, University of Health Sciences, Lahore 42000, Pakistan; 5Department of Animal Breeding & Genetics, University of Veterinary and Animal Sciences, Lahore 42000, Pakistan; 6Services Hospital, Jail Road, Lahore 42000, Pakistan

**Keywords:** cardiac dysfunction, *MYH7*, variants, rs121913642, heart failure (HF)

## Abstract

Cardiac dysfunction accelerates the risk of heart failure, and its pathogenesis involves a complex interaction between genetic and environmental factors. Variations in myosin affect contractile abilities of cardiomyocytes and cause structural and functional abnormalities in myocardium. The study aims to find the association of *MYH7* rs121913642 (c.1594 T>C) and rs121913645 (c.667G>A) variants with cardiac dysfunction in the Punjabi Pakistani population. Patients with heart failure (*n* = 232) and healthy controls (*n* = 205) were enrolled in this study. *MYH7* variant genotyping was performed using tetra ARMS-PCR. *MYH7* rs121913642 TC genotype was significantly more prevalent in the patient group (*p* < 0.001). However, *MYH7* rs121913645 genotype frequencies were not significantly different between the patient and control groups (*p* < 0.666). Regression analysis also revealed that the rs121913642 C allele increases the risk of cardiac failure by ~2 [OR:1.98, CI: 1.31–2.98, *p* < 0.001] in comparison to the T allele. High levels of the cardiac enzymes cardiac troponin I (cTnI) and CK-MB were observed in patients. There was also an increase in total cholesterol, LDL cholesterol, and uric acid in patients compared to the healthy control group (*p* < 0.001). In conclusion, the *MYH7* gene variant rs121913642 is genetically associated with cardiac dysfunction and involved in the pathogenesis of HF.

## 1. Introduction

Cardiomyopathy is a well-known disease of heart muscle and is associated with cardiac dysfunction. The contractile apparatus of normal myocardial cells is organized as a regular array of actin and myosin filaments which are grouped into myofibrils. Each myosin molecule consists of two heavy chains associated with four light chains and exists in various isoforms. Cardiomyopathies are classified into four types, based on their prevalence: (1) hypertrophic cardiomyopathy (HCM), (2) dilated cardiomyopathy (DCM), (3) restrictive cardiomyopathy (RCM), and (4) arrhythmogenic right ventricular dysplasia (ARVD/C) [1,2]. HCM is a common inherited heart failure. It is characterized by the hypertrophy of the left ventricle with the predominant involvement of the interventricular septum (results in diastolic dysfunction), and its prevalence is found to be 1:500 [3,4]. DCM is another major cause of heart failure. It is characterized by conduction defects (atrioventricular block or sinus node dysfunction) and results in systolic dysfunction with reduced ejection fraction [5]. Cardiac dysfunction, both systolic and diastolic, is associated with increased incidence of heart failure.

MYH7 is major gene that encodes the β-myosin heavy-chain (β-MHC) subunit of cardiac myosin. Modifications in myosin directly affect myocardial mechanical function, thus they are considered the cause of dysfunctional myocardial performance leading to heart failure [6]. This is a major cause of morbidity and mortality worldwide. Mutations in MYH7 have been studied and are found to be strongly associated with heart failure. MYH7 variations have been extensively recognized in all regions of β-MHC protein with more frequent occurrence in the ATPase domain, actin-binding domain, and domains responsible for force transmission [7]. Variations in the MYH7gene are predominately missense type, with a dominant negative effect resulting in the formation of mutant peptide. This can be incorporated in sarcomere, which imparts myosin’s motor function [8]. To date, 186 and 73 mutations have been reported for β-MYH7 gene in HCM and DCM cases, respectively [9].

MYH7 rs121913642 (c.1594T>C) variant in exon 16 of the cardiac β-myosin heavy chain causes a Ser532Pro missense mutation [10]. MYH7 rs121913642 causes a change of conserved amino acid serine to proline near the actin-binding domain of myosin [11]. Ser532Pro residue lies in a highly conserved a-helix domain of myosin, which is involved in forming strong hydrophobic interactions between actin and myosin [12]. The MYH7 rs121913645 (c.667G>A) variant in exon 8 of the cardiac β-myosin heavy chain (MYH7), leads to substitution of alanine to threonine at residue 223(Ala223Thr) in the ATP-binding domain of myosin protein, and causes exchange of a non-polar side chain against an uncharged polar group [13]. The Ala223Thr mutation is thought to affect thermostability and protein folding [14]. Hence, the current study aims to find out the prevalence and association of MYH7 rs121913642 and rs121913645 with cardiac dysfunction in the Pakistani Punjabi population. Identification of disease-associated MYH7 variants will lay groundwork to gain deeper insights into its physiological function in order to unravel novel mechanisms of cardiac degeneration.

## 2. Materials and Methods

### 2.1. Study Population

In total, 232 patients with heart failure and 205 healthy controls were enrolled in this study. All patients underwent a physical examination, echocardiography, and ECG at the Wazirabad Institute of Cardiology (WIC), Pakistan. Heart failure patients were ischemic with reduced or preserved ejection fraction, while healthy controls were healthy subjects without ischemic heart disease, cardiac failure, hypertension, diabetes, cancer, or any chronic infectious disease. Patient and control groups were matched for age, gender, and smoking history. Written informed consent was obtained from all of the studied subjects. The study was approved by the Institutional Ethical Committee Board of the University of Sargodha in Sargodha, Pakistan. All protocols and procedures used in this study were according to the Declarations of Helsinki.

### 2.2. Sample Collection and DNA Isolation

Five ml blood was collected from each subject of the study and put in two vacutainers: one for biochemical analysis and the other for genetic analysis. For the biochemical analysis, serum was separated from whole blood by centrifugation at 5000 rpm for 5 min and stored at −20 °C until further analysis. Genomic DNA was extracted from whole blood using the standard phenol-chloroform-isoamyl alcohol DNA-extraction method. Quality of DNA was determined on agarose gel and stored at 4 °C.

### 2.3. Biochemical Analysis

Biochemical analysis of serum for the concentrations of triglycerides, total cholesterol, LDL-C, HDL-C, albumin, total protein, creatinine, urea, and uric acid was carried out by standard kit methods recommended by vendor (Merck Inc., Darmstadt, Germany) on a semi-automated Clinical Chemistry Analyzer Mirolab300 (Merck Inc., Germany). Analysis of clinically important cardiac enzymes, i.e., creatine kinase (CK), CK-MB, and cardiac troponin I (cTnI) were performed at the hospital laboratories.

### 2.4. Genetic Analysis

For genetic variants (rs121913642 and rs121913645) in the MYH7, primers were designed for tetra-primer amplification refractory mutation system polymerase chain reactions (T-ARMS-PCR) using the online software Primer1 (http://primer1.soton.ac.uk/primer1.html 11 June 2018). Primers were optimized by considering melting temperature, hairpin loop, heterodimer, homodimer, and GC content using the online tool OligoAnalyzer (https://eu.idtdna.com/pages/tools/oligoanalyzer 12 June 2018). An in silico amplification of rs121913642 and rs121913645 by T-ARMS-PCR was checked by in silico PCR tool of UCSC genome browser (https://genome.ucsc.edu/cgi-bin/hgPcr 14 June 2018) (Table 1).

PCR reaction conditions for the genotyping of selected MYH7 variants (rs121913642 and rs121913645) were: initial denaturation at 95 °C for 3 min, followed by 30 cycles of 95 °C for 30 s of denaturation, primer annealing at 68.5 °C for 30 s (for rs121913642), and at 62 °C for 45 s (for rs121913645), extension at 72 °C for 30 s, and final extension at 72 °C for 7 min in T100TM thermal cycler (Bio-Rad Laboratories, Inc. Berkeley, CA, USA).

For the genotyping of rs121913642 and rs121913645, PCR reaction mixtures of total volume 25 μL each were prepared in 200 μL PCR tube. PCR reaction mixtures for both variants had the same amounts of genomic DNA (100–200 ng/μL), 10XPCR buffer [1.5 μL (750 mMTris-HCL, pH 8.8)], and 2 mMdNTPs (2 μL), while the amounts of MgCl2, primers and Taq polymerase were different for rs121913642 and rs121913645. For rs121913642, 1.8 μL of 25 mM MgCl2, 0.5 μL of each primer, and 1.6U Taq polymerase were used. However, for rs121913645, 2 μL of 25 mM MgCl2, 0.5 μL of forward outer, reverse outer, and reverse inner primers, and 0.7 μL of forward inner primer and 2.5 U Taq polymerase were added to the reaction mixture. DdH2O was also added to make the total volume of 25 μL. The amplified products of both variants were run on 2% agarose gel to which ethidium bromide was already added. A 1 kb ladder was run along the samples for size estimation and the gel was visualized under UV light in Gel DocTM EZ System (Bio-Rad Laboratories, Inc.).

Genotyping of MYH7 rs121913642 and rs121913645 was conducted for all of the subjects of this study. Tetra ARMS-PCR analysis of rs121913642 revealed a control band at 627 bp, T allele band at 374 bp, and C allele band at 312 bp. Likewise, rs121913645 analysis showed amplification of 653 bp (control band), 419 bp (G allele), and 287 bp (A allele) (Figure 1).

### 2.5. Statistical Analysis

Data analyses were performed by using SPSS version 20 (IBM Inc. Armonk, NY, USA). Student’s *t*-test was applied for comparison of clinical and biochemical parameters (as Mean ± S.D) of the study groups, i.e., patients and healthy controls. A gene-counting method was used to calculate the allelic frequencies while a chi-square (χ^2^) test was used to test for the association of genotypes and alleles with cardiac dysfunction. ANOVA was used to see the association of clinical and biochemical parameters with MYH7 polymorphisms in the study population. Moreover, logistic regression analysis was performed to find the disease risk associated with alleles.

## 3. Results

### 3.1. Clinical and Biochemical Analysis

There was no significant difference in age, gender, or smoking frequency between patients and controls. Descriptive analysis of patients (*n* = 232) and controls (*n* = 205) revealed that systolic and diastolic blood pressure was significantly higher in the patient group compared to the control group. Ejection fraction (EF) was significantly reduced in patients compared to healthy controls (*p* < 0.001). Serum concentration of cardiac troponin I (cTnI) was higher in patients, but difference was not statistically significant (*p* = 0.067). All CK measurements in patients were significantly higher than normal. Furthermore, the concentration of total cholesterol, LDL cholesterol, uric acid, urea, creatinine, and albumin were also significantly higher in the patient group as compared to the control group. However, the concentration of HDL cholesterol was less in the patient group (40 ± 12 vs. 48 ± 7 mg/dL); however, this difference was not statistically significant (*p* = 0.055) (Table 2).

### 3.2. Genetic Analysis

Frequency of the MYH7 rs121913642 TT genotype was significantly higher in control subjects (98.5% vs. 96.6%), while the frequency of the TC genotype was higher in patients (1.5% vs. 3.4%). Surprisingly, the CC genotype was not observed in any subject and the Fisher exact test revealed that the genotypic and allelic frequencies of rs121913642 were significantly different among the patient and control groups (*p* < 0.001). Similarly, genotypic analysis of MYH7 rs121913645 also revealed that the patient group has higher frequency of GA (1.3% vs. 1%; *p* = 0.666), while the frequency of homozygous normal genotype carriers is more in the control group (99% vs. 98.7%; *p* = 0.677). There are differences in the genotypic and allelic frequencies of rs121913645; however, these are not significantly different (*p* = 0.666 and *p* = 0.677) (Table 3).

Although *MYH7* rs121913645 GA genotype carriers also showed significantly reduced ejection fraction (*p* = 0.001), the concentration of other cardiac parameters, such as creatine kinase, CK-MB, and cardiac troponin I, were not significantly associated with the selected *MYH7* rs121913645 GA genotype. Apart from the disease-associated parameters, some other clinically important biochemical parameters such as LDL cholesterol and creatinine also showed significant difference between genotypes (homozygous major and heterozygous) in the patient group. One-way ANOVA analysis demonstrated that the concentration of LDL cholesterol (*p* = 0.011) and creatinine (*p* = 0.046) was significantly higher in MYH7 rs121913645 GA genotype carriers (Table 4).

One-way ANOVA analysis revealed that the genotypes of MYH7 rs121913642 significantly associated with serum concentration of CK-MB and cardiac troponin I (cTnI). The concentration of CK-MB was raised in patients carrying TC genotype (*p* = 0.001), while the same genotype carriers also presented elevated levels of cTnI. Cardiac ejection fraction (EF) was significantly reduced in patients carrying the heterozygous genotypes (MYH7 rs121913642 TC and MYH7 rs121913645 GA). The disease-associated parameters were showing statistically significant pathological changes in the carriers of polymorphic alleles. Hence, these results can provide an insight to the association of minor allele with disease onset and severity. Logistic regression analysis complemented these assumptions by statistically showing the association of MYH7 rs121913642 C allele with the disease onset. Logistic regression analysis showed that the MYH7 rs121913642 C allele (minor allele) increases the risk of cardiac dysfunction by ~2-fold in comparison to the rs121913642 T allele (major allele) [OR 1.98, CI: 1.317–2.982, *p* < 0.001] (Table 5).

## 4. Discussion

A major finding of the current study is the association of *MYH7* rs121913642 TC genotype and *MYH7* rs121913642 C allele with the cardiac dysfunction. Moreover, frequency of minor alleles carrying genotypes (for rs121913642 and rs121913645) were unexpectedly higher in Punjabi Pakistani population. Detailed results of the current study demonstrated higher frequency of *MYH7* rs121913642 TC in patients with heart failure. The *MYH7* rs121913642 TC genotype also showed an association with the serum concentrations of cardiac biomarkers CK-MB and cardiac troponin I (cTnI). As MYH7 rs121913642 CC (homozygous minor genotype) was not observed in any individual of the study cohort, allelic analysis was carried out to check the influence of alleles on disease outcome. Allelic study showed that *MYH7* rs121913642 C allele was higher in patients than in healthy controls. Regression analysis also favored these findings by showing that the carriers of *MYH7* rs121913642 C allele were at ~2 folds higher risk of heart failure compared to *MYH7* rs121913642 T allele carriers. Descriptive analysis also revealed the association of *MYH7* rs121913645 GA genotype with cardiac risk factors including LDL cholesterol and creatinine. However, no significant association was observed between the genotypes and alleles of *MYH7* rs121913645 with cardiac dysfunction.

Cardiac dysfunction due to abnormal heart muscle is known as cardiomyopathy. Dilated cardiomyopathy (DCM) is characterized by left ventricle dilation and systolic dysfunction, and is the most severe and prevalent form of cardiomyopathy [15]. DCM often leads to heart failure. Current genetic studies of various single gene mutations have provided significant insight into the complex progression of DCM into heart failure. So far, over 40 genes have been demonstrated to contribute to dilated cardiomyopathy. The respective gene products can be classified into several functional groups, such as sarcomere proteins, structural proteins, ion channels, and nuclear envelope proteins [16].

Dilated cardiomyopathy (DCM) and hypertrophic cardiomyopathy (HCM) are important cause of arrhythmias, heart failure, and sudden cardiac death. Mutations in sarcomere proteins, including cardiac β-myosin heavy chain (β-MHC), cause both diseases [13]. Different MYH7 mutations gave enhanced or reduced rates of ATP binding, ATP hydrolysis, ADP release, or exhibited altered ATP, ADP, or actin affinity [17]. DCM-causing mutations of MYH7gene resulted in systolic dysfunction, although left ventricular size and ejection fraction remained constant. In contrast, preserved systolic function is the predominant early manifestation of MYH7 mutations that lead to HCM [18].

*MYH7* variants have been linked to cardiac dysfunction in various world populations. Cardiac β-myosin heavy-chain (β-MHC) mutations affect the mechanical properties of myosin, which are the molecular motors of the heart. MYH7 rs121913642 variant causes a change of conserved amino acid serine to proline (Ser532Pro) at residue number 532 near the actin binding site of cardiac β-myosin heavy chain (β-MHC), which has shown functional consequences. MYH7 rs121913645 results in a nucleotide substitution G to A, causing a change of alanine by threonine (Ala223Thr) at residue 223 near the ATP binding site in the upper 50 kD of the β-myosin heavy chain (β-MHC).

In the current study, interestingly, homozygous minor genotypes (rs121913642 CC and rs121913645 AA) were not observed in any individual of the control and patient groups. This can be due to the selection pressure during embryonic development against the homozygous minor genotypes (rs121913642 CC and rs121913645 AA), leading to complete loss of cardiac function and embryo death. A well-known database of genomic variations, ClinVar, reported that rs121913642 is a pathogenic variation, while rs121913645 has conflicting calls of pathogenic and uncertain significances. However, rs121913642 was not found in reference database GnomAD, while rs121913645 appears once in GnomAD. Although previous literature and variant databases such as Ensembl and dbSNP show that the minor allele carrying genotypes of selected MYH7 variants are extremely rare and have not been observed in neighboring populations of Pakistan, the current study showed a significantly higher frequency of heterozygous genotypes. This difference in the minor allele frequency between our Punjabi Pakistani population and other populations could be due to founder effects such as high rate of consanguineous marriages. Moreover, this population has also encountered some bottleneck events, such as migration during the independence movement on 14 August 1947, and wars in 1965 and 1971. These events reduced the population size (loss of genetic diversity) and encouraged repeated marriages of close relatives.

Point mutations in cardiac myosin, the heart’s molecular motor, produce distinct clinical phenotypes in hypertrophic and dilated cardiomyopathy [19]. A genome-wide family-based linkage study conducted in the Brazilian population reported the association of *MYH7* rs121913642 with dilated cardiomyopathy. Ser532Pro mutation causes ventricular dilation and reduced contractile function, which eventually leads to heart failure [9]. A study from the German population found the association of *MYH7* rs121913645 with dilated cardiomyopathy. Mutation screening analysis of MYH7 in DCM patients revealed that Ala223Thr affects protein folding and thermostability [10]. A functional study in murine model demonstrated that *MYH7* mutations caused changes in actin–myosin cross bridge kinetics. *MYH7* missense variants (Phe764Lys and Ser532Pro) elevate rates of force development and Mg-ATP binding, a feature of cardiac myofilaments which underlie the development of dilated cardiomyopathy [20]. Schmitt et al. [21] also found that *MYH7* variants in mouse models cause a loss of contractile function and early-onset left-ventricular dilation, which are characteristics of DCM consistency with the human phenotypes [22]. These findings showed that HCM and DCM occur due to alternations in the fundamental mechanical properties of myosin, which are further affected by the variations in the *MYH7*, and lead to heart failure. The current study also indicated that *MYH7* variant rs121913642 (Ser532Pro) increases the risk of heart failure in patients from Punjab, Pakistan. However, *MYH7* rs121913645 did not show any association with cardiac failure though its genotypes and had an influence on some of the clinically important biochemical parameters.

## 5. Conclusions

In conclusion, the current study highlights the difference in population frequency of *MYH7* variants (rs121913642 and rs121913645) in Punjabi Pakistani population. The rs121913642 C allele appears to be a common variant and endemic in this population. Moreover, the *MYH7* rs121913642 TC genotype increases the risk of heart failure and is significantly associated with serum concentrations of cTnI and CK-MB. Another genetic variant, rs131913645, is significantly associated with serum LDL cholesterol and creatinine levels. However, it is the first study of Punjabi Pakistani population for these two MYH7 gene variants, and this warrants replication in other ethnic groups of the Pakistani population for these and additional genetic variants in cardiac-related genes to obtain clinically useful insights. Furthermore, disease penetrance of these variants and identification of disease modifiers can be assessed by family-based linkage studies.

## Figures and Tables

**Figure 1 genes-13-01554-f001:**
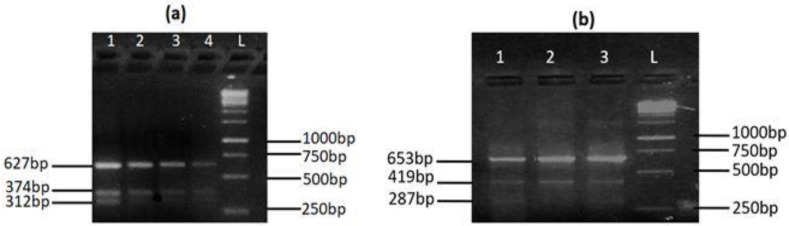
Tetra ARMS-PCR assay results for the genotypes of (**a**) *MYH7* rs121913642 and (**b**) *MYH7* rs121913645.

**Table 1 genes-13-01554-t001:** Tetra ARMS-PCR primer sequence details for genotyping of MYH7.

Primers	Primers Sequences (5′–3′)	T_m_ (°C)	Product Size
*MYH7*rs121913642			
Forward outer	CCTAGCATCTCAGGCATCTGGGTCGTGGAGTG	66.8	Control = 627 bp T allele = 374 bp C allele = 312 bp
Reverse outer	CCTTGGCAGAAACCCTGCTCCTCTGTACCG	66.5	
Forward inner	CCTGCTTCCTCAGCACATGGGCATCAGGT	67.1	
Reverse inner	CCTTGGGGAACATGCAGTCCTCTTCCAGGATTGG	66.9	
*MYH7*rs121913645			
Forward outer	CAGCCGTGACCTCTCTGCATCAGAAGACAG	64.7	Control = 653 bp A allele = 287 bp G allele = 419 bp
Reverse outer	CTGAGCCTAGCAGATTCATGGCACTCACAGG	64.6	
Forward inner	GCACGCTTGAGGACCAGATCATCCCGA	65.5	
Reverse inner	GCCAAATGCCTCCAGAGCAGGGTTAGC	65.2	

**Table 2 genes-13-01554-t002:** Comparison of clinical and biochemical parameters in patients and healthy subjects.

Parameter	Control (*n* = 205)	Patient (*n* = 232)	Significance *(p-*Value)
Systolic BP (mmHg)	118 ± 15	140 ± 12	<0.001
Diastolic BP (mmHg)	79 ± 11	88 ± 15	<0.001
Ejection Fraction % (EF)	55 ± 5	45 ± 10	<0.001
Creatine kinase (CK) (U/L)	62 ± 41	149 ± 61	<0.001
CK-MB (ng/mL)	3.7 ± 1.1	5.3 ± 2.4	<0.001
Cardiac Troponin I (cTnI) (ng/mL)	0.02 ± 0.01	0.05 ± 0.02	0.067
Cholesterol (mg/dL)	185 ± 40	221 ± 50	<0.001
HDL (mg/dL)	48 ± 7	40 ± 12	0.055
LDL (mg/dL)	83 ± 11	126 ± 40	<0.001
Triglycerides (mg/dL)	275 ± 169	239 ± 159	0.048
Serum Uric acid (mg/dL)	5.9 ± 1.8	7.1 ± 2.7	<0.001
Urea (mg/dL)	16 ± 10	38 ± 10	*<*0.001
Creatinine (mg/dL)	0.7 ± 0.2	1.1 ± 0.8	*<*0.001

**Table 3 genes-13-01554-t003:** Genotypic and allelic frequencies of *MYH7* rs121913642 and rs121913645.

Variant		Control (*n* = 205)	Patient (*n* = 232)	Significance (*p*-Value)
rs121913642 (T>C)	TT	202 (98.5%)	224 (96.6%)	χ2 = 12.35 *p* < 0.001
	TC	3 (1.5%)	8 (3.4%)	
	CC	0 (0%)	0 (0%)	
	T	407 (99.3%)	456 (98.2%)	χ2 = 10.40 *p* < 0.001
	C	3 (0.7%)	8 (1.8%)	
rs121913645 (G>A)	GG	203 (99%)	229 (98.7%)	χ2 = 0.141 *p* = 0.666
	GA	2 (1%)	3 (1.3%)	
	AA	0 (0%)	0 (0%)	
	G	408 (99.5%)	461 (99.3%)	χ2 = 0.106 *p* = 0.677
	A	2 (0.5%)	3 (0.7%)	

**Table 4 genes-13-01554-t004:** Analysis of clinical and biochemical parameters among various genotypes of *MYH7* rs121913642 and rs121913645 in cardiac patients.

Parameter	*MYH7* rs121913642			*MYH7* rs121913645		
	TT	TC	*p-*Value	GG	GA	*p-*Value
Systolic BP (mmHg)	141 ± 18	143 ± 18	0.605	142 ± 18	136 ± 18	0.142
Diastolic BP (mmHg)	80 ± 10	82 ± 10	0.346	82 ± 10	79 ± 9	0.210
Ejection fraction (%)	44 ± 10	39 ± 11	**0.013 ***	43 ± 5	40 ± 11	**0.001 ***
Creatine kinase (CK) (U/L)	148 ± 61	149 ± 60	0.930	145 ± 60	147 ± 60	0.873
CK-MB (ng/mL)	5.1 ± 2.4	5.5 ± 2.9	**0.001 ***	5.3 ± 2.0	5.3 ± 2.4	0.763
Cardiac troponin I (cTnI) (ng/mL)	0.05 ± 0.02	0.06 ± 0.03	**0.007 ***	0.05 ± 0.02	0.05 ± 0.01	0.757
Cholesterol (mg/dL)	226 ± 50	212 ± 45	0.095	222 ± 60	212 ± 45	0.082
HDL Cholesterol (mg/dL)	64 ± 58	53 ± 18	0.161	60 ± 40	71 ± 50	0.474
LDL cholesterol (mg/dL)	132 ± 80	112 ± 57	0.104	122 ± 58	148 ± 130	**0.011 ***
Triglycerides (mg/dL)	245 ± 164	229 ± 119	0.561	238 ± 144	254 ± 187	0.997
Urea (mg/dL)	43 ± 19	41 ± 20	0.566	41 ± 19	46 ± 18	0.869
Uric acid (mg/dL)	7.1 ± 2.7	7.1 ± 2.5	0.882	7.1 ± 2.6	7.1 ± 3.1	0.221
Creatinine(mg/dL)	1.1 ± 0.6	1.1 ± 0.5	0.978	1.0 ± 0.4	1.3 ± 0.6	**0.046 ***

* *p*-value is significant.

**Table 5 genes-13-01554-t005:** Regression analysis of *MYH7* rs121913642 and rs121913645 for disease risk.

Variant	Allele	Allele	Variant	Allele	Allele
rs121913642	T	C	rs121913645	G	A
	Ref.	1.982 (1.317–2.982) *p* < 0.001 *		Ref.	0.878 (0.506–1.522) *p* = 0.642

* *p*-value is significant.

## Data Availability

The data supporting the conclusions of this article are available in the text of this article.

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
