# Peer review of "Genetic Association of Beta-Myosin Heavy-Chain Gene (MYH7) with Cardiac Dysfunction"

_genes, 2022, doi:10.3390/genes13091554_

Round 1

Reviewer 1 Report

Summary

The authors performed genotyping of 2 genetic variants in MYH7 that have previously associated with cardiomyopathy in a case-control heart failure cohort from the Punjabi Pakistani population. They found that these variants were not rare in the Punjabi Pakistani population but were either of low frequency or common. There was clear evidence of phenotype-genotype association for the variant currently classified as pathogenic on ClinVar. However, phenotype-genotype association for the variants classified as of uncertain significance was less clear.

General Impression

  1. Manuscript is not well written, i.e. sections of the introduction and discussion could have been organized better to support the rationale for this study
  2. Authors do not appear to be familiar with the language in genetics and genetic databases and have not cited relevant or correct information
  3. The novelty of this study is that the Ser532Pro is rather prevalent in the Punjabi Pakistani population and will require significant re-write to get all this point across

Introduction

  1. “Autosomal dominant mutations in the MYH7 and MYBPC3 account for nearly 80% of inherited HCM” Citation does not support this assertion. In addition, 80% seems high among inherited HCM.
  2. “Functional studies conducted in on different world populations revealed the pathogenic effect of MYH7 rs121913642 (Ser532Pro) and rs121913645 (Ala223Thr) missense variants on the structure and function of myosin.” Please provide citation. Also, these were not functional studies. The first were family-based studies. The second was case-control resequencing.
  3. While Ser532Pro has been classified as pathogenic, variant classification for Ala223Thr showed conflicting calls and was most recently classified as a variant of uncertain significance. There the authors should not refer to them both as mutations.
  4. Were there unpublished data on the allele frequencies of these 2 variants in the Punjabi Pakistani population? If yes, please mention them. If not, the authors should make a stronger argument on why they chose to look at these 2 particular variants, although I do find it intriguing why the prevalence of the rs121913642, which is a bono fide pathogenic variant, is so high in the Punjabi Pakistani population.

Results

  1. Authors should specify how heart failure patients were recruited, whether they were ischemic, non-ischemic, reduced or preserved ejection fraction, and show clinical characteristics supporting their assertion in the first sentence that there is “no significant difference in age, gender and smoking frequency between patients and controls.” Where is Table 2?
  2. “Similarly genotypic analysis… however, these are not significantly different.” Should remove any assertion that there is any difference in the first sentence, since there is no statistical evidence that there is any. Where is Table 3?
  3. Were the individuals with elevated LDL and Cr individuals with atherosclerotic disease?
  4. Did any of the genotype positive individuals in the control population show subclinical finding of cardiomyopathy, such as enlarged chamber or reduced ejection fraction, while remaining symptom free?

Discussion

  1. It is a novel finding that the population frequency of these 2 variants are this high in the Punjabi Pakistani population. Please emphasize this point.
  2. Were there any individual harboring both missense variants?
  3. There is possible selection process during embryonic development against those homozygous for the minor allele for both genes.
  4. “A genome-wide association-study conducted in Brazilian…” This is incorrect. It was a genome-wide family-based linkage study and not a genome-wide association study.
  5. It is intriguing that there were association of the Ala223Thr variant with LDL and Cr, suggesting interaction with other unassessed atherosclerotic or metabolic processes.

The authors should further delineate in the results and suggest possible mechanistic link in the discussion.

Conclusion

  1. The use of genotypic spectrum seems inappropriate. Please reword.
  2. Consider citing IndGen Genome https://indigen.igib.in and stating the importance of population genetic databases in understanding mutation prevalence and disease risk
  3. Future directions may involve addressing disease penetrance of these variants and identification of disease modifiers through family-based linkage studies.

Author Response

The annotated point-by-point response to the reviewer comments has been attached as a pdf file "Reviewer.docx_annotated_reply.pdf". Please

Reviewer 2 Report

This is a novel and interesting study with groups of patients with heart failure (n=232) and healthy controls (n=205). The study investigated the association of MYH7 rs121913642 (c.1594T>C) and rs121913645 (c.667G>A) variants with cardiac dysfunction in Punjabi Pakistani population. They found that MYH7 gene variant rs121913642 was genetically associated with cardiac dysfunction and involved in the pathogenesis of heart failure. In the manuscsript, Tables 2-5 were missing.  Figure 1 should be appropriately made .

Author Response

We thank the reviewer for their valuable and supportive comments and suggestions. We have revised the manuscript accordingly. Please

Round 2

Reviewer 1 Report

The authors have made significant improvement in the presentation of their data and findings. Please rearrange paragraphs between introduction and discussion for a more logical flow of ideas. Please state clearly that p.Ser532Pro has been reported to be pathogenic in ClinVar and not found in reference databases such as Gnomad and p.Ala223Thr has conflicting calls of pathogenic and uncertain significance in ClinVar and found once in Gnomad. Again, please state how control patients were selected. The observed associations depend on how the heart failure and control patients were selected. Details of this selection will help the reader make sense of the observed associations. Specifically, are controls ‘healthy’ controls or individuals with coronary disease yet have heart failure symptoms? Were they matched for age, gender and smoking history? On page 4 “One-way ANOVA analysis demonstrated that the MYH7 rs121913645 GA genotype significantly influence the concentration of LDL-cholesterol ….” This statement implies causation. Please restate the relationship as an association instead. Novel conclusion from this study: 1 (as stated by the authors already): rs121913642 C allele appears to be a common variant  p.Ser532Pro appears to be endemic in this Pakistani population. 2. If the patient group is truly an ischemic heart failure cohort and control an ischemic non-heart failure cohort, a pathogenic MYH7 variant can be conceptualized as a modifier of ischemic heart disease, and this would be a novel observation.

Author Response

Respected Reviewer,

Manuscript title: Genetic association of Beta-Myosin Heavy Chain (MYH7) gene with Cardiac dysfunction

(Manuscript ID: genes-1626353)

Response to reviewer comments

Comment No.1. Please rearrange paragraphs between introduction and discussion for a more logical flow of ideas.

Answer: Introduction and discussion has been improved as per kind suggestions of reviewer.

Comment No. 2: Please state clearly that p.Ser 532Pro has been reported to be pathogenic in ClinVar and not found in reference database such as Gnomad and p.Ala223Thr has conflicting calls of pathogenic and uncertain significance in ClinVar and found once in Gnomad.

Answer: Mentioned information about MYH7 gene variations and genome databases has been added in discussion section. (Page 8, line 239-242: highlighted in yellow)

Comment No.3: Please state clearly are controls ‘healthy’ controls or individuals with the coronary disease yet have heart failure symptoms? Were they matched for age, gender and smoking history?

Answer: Healthy controls were healthy subjects without coronary disease, ischemic heart disease, cardiac failure, hypertension, diabetes, cancer or any chronic infectious disease. Moreover, patient and control groups were matched for age, gender and smoking history. (Page 2, line 70-73: highlighted in yellow)

Comment No.4: On page 4 “One-way ANOVA analysis demonstrated that the MYH7 rs121913645 GA genotype significantly influences the concentration of LDL-cholesterol” This statement implies causation. Please restate the relationship as an association instead.

Answer: The said line on page 4 has been rephrased. (Page 5, lines 166-168: highlighted in yellow)

Comment No. 5: Novel conclusion from this study: 1 (as stated by the authors already): rs121913642 C allele appears to be a common variant p. Ser532Pro appears to be endemic in this Pakistani population. 2. If the patient group is truly an ischemic heart failure cohort and controls an ischemic non-heart failure cohort, a pathogenic MYH7 variant can be conceptualized as a modifier of ischemic heart disease, and this would be a novel observation.

Answer: Current study is the first report from Punjabi Pakistani about the association of MYH7 gene variations with ischemic heart failure in this population. Interestingly, the collected data indicates that the frequency of the rs121913642 C allele is higher in our population as compared to other populations. Although the patient cohort was truly ischemic heart failure, however, the control group was not an ischemic non-heart failure cohort instead they were healthy subjects without coronary disease, ischemic heart disease or heart failure. Thus, the observation that rs121913642 is a modifier of ischemic heart disease needs replicative studies with more strict selection criteria for patients and control. The first conclusion has been included in the conclusion section. (Page 9, line 277: highlighted in yellow).

Reviewer 2 Report

No further comments.

Author Response

Respected Reviewer,

On the behalf of my co-authors, I really appreciate your time and efforts to review this manuscript and help us to add your valuable comments and suggestions for the shape of this paper. 

Regards